# Allergic Rhinitis as an Independent Risk Factor for Postoperative Recurrence of Children Chronic Sinusitis

**DOI:** 10.3390/children10071207

**Published:** 2023-07-12

**Authors:** Caixia Zhang, Hua Zhang, Qingping Tang, Junyi Zhang, Shuo Wang, Zhihai Xie, Weihong Jiang

**Affiliations:** 1Department of Otolaryngology Head and Neck Surgery, Xiangya Hospital of Central South University, No. 87 Xiangya Road, Kaifu District, Changsha 410008, China; 218102155@csu.edu.cn (C.Z.); entxy@126.com (H.Z.); zjy1857@126.com (J.Z.); xiedoctor@csu.edu.cn (Z.X.); 2Hunan Province Key Laboratory of Otolaryngology Critical Diseases, Xiangya Hospital of Central South University, Changsha 410008, China; 3National Clinical Research Center for Geriatric Disorders, Xiangya Hospital of Central South University, Changsha 410008, China; 4Anatomy Laboratory of Division of Nose and Cranial Base, Clinical Anatomy Center of Xiangya Hospital, Central South University, Changsha 410008, China; 5Department of Rehabilitation, Brain Hospital of Hunan Province, Hunan University of Chinese Medicine, Changsha 410008, China; tqingping1111@126.com; 6Department of Pediatric, Xiangya Hospital, Central South University, Changsha 410008, China; wangshuo998@sina.com

**Keywords:** chronic sinusitis, allergic rhinitis, recurrence, risk factor, children

## Abstract

(1) Background: The recurrence rate of childhood recurrent sinusitis varies widely between 12% and 50%, with the postoperative recurrence risk factors remaining largely unidentified. We sought to enhance the understanding of chronic rhinosinusitis (CRS) via a retrospective observational childhood cohort. (2) Methods: The study recruited 125 cases. Demographic data and univariate and multivariate logistic regression analyses were conducted to investigate potential risk factors of childhood recurrent sinusitis following functional endoscopic sinus surgery (FESS). (3) Results: A postoperative recurrence rate of 21.6% was determined. Among the participants, 21 cases presented a history of allergic rhinitis (AR), with the remaining 104 cases being AR-free. A significantly heightened recurrence rate was noted in those bearing a history of AR compared to their counterparts devoid of such history (*p* < 0.000). The fully adjusted logistic regression model indicated a 21.04-fold increased risk of postoperative recurrence in childhood CRS bearing a history of AR compared to those without an AR history (*p* = 0.000), highlighting the history of AR as an independent risk factor for postoperative childhood recurrent sinusitis (*p* = 0.001); (4) Conclusions: The data implicate AR as an independent risk factor for postoperative childhood recurrent sinusitis.

## 1. Introduction

Chronic rhinosinusitis (CRS) in children is an inflammatory disease that affects the mucosa of the nose and paranasal sinuses with a reported prevalence of about 4% in the pediatric population [1,2]. This prevalent disease often eludes diagnosis in otolaryngology departments, and its gradually increasing prevalence has attracted significant attention [3]. Similar to adult chronic rhinosinusitis, the clinical diagnosis and treatment of CRS in children has evolved from mainly relying on doctors’ personal experience to being based on evidence-supported clinical guidelines. Accurate diagnostic and treatment strategies should be developed based on the etiology of CRS in children [1]. The clinical manifestations of pediatric CRS are nonspecific, including symptoms like nasal congestion, headaches, facial pain/pressure, cough, purulent nasal discharge and olfactory dysfunction, persisting for more than 12 weeks. This is often accompanied by other symptoms such as memory loss and lethargy, significantly impacting children’s growth and education, affecting family life quality, and imposing a substantial economic burden [3,4].

According to the clinical phenotype of polyps, CRS can be classified into two categories: CRS without nasal polyps (CRSsNP) and CRS with nasal polyps (CRSwNP). The diagnosis primarily relies on clinical history and examination procedures such as nasal endoscopy and computed tomography (CT). The initial medical management of CRS in children chiefly involves nasal saline irrigation, nasal saline spray and oral antibiotics [3]. Recently, an expanding corpus of evidence has begun advocating for surgical intervention in pediatric patients who exhibit limited response to conventional medical management [3,5,6]. FESS not only assuages symptomatic distress but also enhances the quality of life in a majority of patients. However, despite the employment of standard pharmacological treatments (oral antibiotics, nasal saline sprays and intranasal steroid sprays) and/or functional endoscopic sinus surgery (FESS), a subset of pediatric patients continues to experience persistent symptoms or swift recurrence. Certain researchers have reported that the postoperative recurrence rate of CRS ranged from 12% to 50% during the follow-up period after surgery, posing a formidable challenge in disease management [7].

CRS’s histopathology is highly heterogeneous. Recently, it is mainly characterized by local dominant infiltrating immune cell types, such as eosinophil type with type 2 inflammatory reaction and lymphocyte/plasma cell type with non-2 inflammatory reaction. Studies have shown that the intrinsic type of CRSwNP is related to the clinical symptoms of the disease, which has certain guiding significance in the selection of treatment schemes: type 2 inflammation is often accompanied by allergic diseases such as asthma, which is related to the recurrence of the disease [8,9]. Currently, the majority of the literature has reported on the prognosis evaluation, risk factors, or mechanisms of CRS in adults or the general population [10,11]. However, the understanding related to the recurrence of CRS in children remains vague. Children, representing the future of our society, should receive greater attention and be studied as a distinct subgroup. Inflammatory diseases of nasal and paranasal mucosa, as components of the upper airway in children, have high prevalence and can cause significant harm [12], underscoring the urgency for diligent diagnosis and therapeutic strategies. Allergic rhinitis (AR) has a close relationship with chronic sinusitis, and inflammatory diseases of the upper airway in children often coexist [12,13]. A myopic focus on a singular disease’s treatment may fail to yield an efficacious control or resolution [12,13]. The objective of this study is to delve into potential risk factors, such as complications and clinical blood biochemical examination findings, for recurrence post-FESS in children suffering from CRS unresponsive to conservative drug therapies. This allows for a more profound understanding of CRS. An exploration into these risk factors will facilitate prospective intervention and enhance clinical management of CRS in the pediatric population.

## 2. Materials and Methods

### 2.1. Study Population

A retrospective cohort study was conducted via telephone, establishing a clinical data database for patients diagnosed with CRS as per the European Position Paper on Rhinosinusitis and Nasal Polyps 2012 [14] (EPOS 2012) and the Clinical Guidelines on Chronic Rhinosinusitis in Children [5]. These patients had undergone FESS at Xiangya Hospital of Central South University between May 2017 and May 2022.

Children less than 18 years (*n* = 125) were recruited for the study. These children had suffered from severe symptoms such as severe headache and sleep disruption and had been treated with standardized treatment strategies (oral antibiotics, nasal saline sprays, and intranasal steroid sprays) in strict accordance with the guidelines before admission, but the therapeutic effect was notably insufficient. Recurrence was identified as the re-emergence of clinical symptoms, and endoscopic and CT evidence for at least 2 months despite a rescue regimen of antibiotics and oral steroids after FESS [10,15]. The diagnosis of AR is based on clinical history, skin tests, or the presence of serum-specific immunoglobulin E (IgE) antibodies to allergens [16]. The diagnosis of asthma relies on the recognition of a characteristic pattern of symptoms, the occurrence of asthma attacks, and sometimes by basic lung function tests [17]. All children with CRS underwent detailed routine otolaryngology, physical and laboratory tests (Figure 1).

This study was conducted by the Declaration of Helsinki and was approved by the Ethics Committee of Xiangya Hospital of Central South University (protocol code: 202306129; date of approval: 19 June 2023). Written informed consent to participate in this study was provided by the legal guardian or close relative of the participant.

For any further inquiries, please feel free to contact corresponding author at any time.

### 2.2. Inclusion and Exclusion Criteria

In line with the published guideline [5,14], children aged less than 18 years who were diagnosed with CRS, underwent FESS at our medical center and had complete clinical data and a follow-up period of at least 1 year were included in the study.

Exclusion criteria included children with incomplete data or those accompanied by fungal sinusitis, allergic fungal rhinosinusitis, benign or malignant sinus tumors, current acute inflammation, a history of radiotherapy, cystic fibrosis, autoimmune diseases, posterior nostril polyps, those aged 18 years or older and children who refused initial or modified surgery.

### 2.3. Variables

The following variables were recorded for all children: sex, age, height, weight, systolic blood pressure, diastolic blood pressure, course of the disease, history of AR, history of asthma, serum total protein, serum albumin, serum globulin, leukocyte ratio, serum total bilirubin, serum direct bilirubin, albumin-globulin ratio, serum total bile acid, serum alanine aminotransferase, serum aspartate aminotransferase, urea, creatinine, uric acid, glucose, glycosylated hemoglobin, triglyceride, cholesterol, white blood cell count, platelet count, neutrophil count, lymphocyte count, eosinophil count, basophil ratio, monocyte ratio, neutrophil ratio, lymphocyte ratio, basophil ratio, and monocyte ratio.

### 2.4. Statistical Analysis

Children were divided into the primary CRS group and the recurrent CRS group. If the continuous variable followed a normal distribution, it was expressed as mean ± SD; otherwise, it was reported as median and interquartile ranges (25%, 75%). Categorical variables were expressed as frequency or percentage. The χ^2^ test (for categorical variables), Student’s *t*-test (for continuous variables with normal distribution), or Mann-Whitney U test (for continuous variables with skewed distribution) were employed to analyze differences between groups. Logistic regression was used to analyze the influence of each variable affecting the postoperative recurrence of sinusitis in children. Multiple binary logistic regression was used to analyze the possible association between AR history and childhood recurrent sinusitis, and three models were constructed to illustrate the stability of the influencing ability of each variable in this relationship: Model 1 adjusted for none; Model 2 adjusted for sex, age, height, and weight; Model 3 adjusted for sex, age, height, weight, serum total protein, serum globulin, albumin-globulin ratio, white blood cell count, neutrophil count, lymphocyte count, monocyte count, basophil ratio and eosinophil ratio. The significance level was set as a two-tailed *p* < 0.05, which indicated that the difference was statistically significant. All analyses were performed using SPSS 22.0 (IBM, Chicago, IL, USA).

## 3. Results

### 3.1. Population Characteristics and Baseline Data of All Participants

A total of 125 children were included in the final data analysis after screening. The longest follow-up period of the study was 5 years, while the shortest was 1 year, with an average follow-up of 3 years. The cohort comprised 98 cases of initial CRS and 27 cases of recurrent CRS. Of these, three patients with CRSsNP were all 17 years old. Among them, one patient with a history of AR experienced recurrence post-surgery and underwent a second operation (Table 1).

In our cohort, there were 21 cases of AR and 8 cases of asthma. The postoperative recurrence rate for children with a history of AR was significantly higher than that for children without AR [13 (48.51) vs. 8 (8.16), *p* < 0.000]. Similarly, the postoperative recurrence rate for children with a history of asthma was higher than that for children without a history of asthma [5 (18.52) vs. 3 (3.06), *p* = 0.012].

Compared with those with initial CRS, the systolic blood pressure levels were relatively lower in children with recurrent CRS [(117.88 ± 13.71) vs. (112 ± 10.37), *p* = 0.041]. Furthermore, serum total protein levels were lower in recurrent CRS compared to initial CRS [68.90 (66.20–72.25) vs. 66.70 (65.20–70.50), *p* = 0.024]. Recurrent CRS had higher serum triglyceride levels compared to primary [0.96 (0.73–1.14) vs. 1.12 (0.88–1.59), *p* = 0.028]. However, no statistical difference was found between the two groups regarding sex, age, height, body weight, diastolic blood pressure, course of the disease, albumin, globulin, albumin-globulin ratio, leukocyte ratio, total bilirubin, direct bilirubin, total bile acid, alanine aminotransferase, aspartate aminotransferase, urea, creatinine, uric acid, glucose, glycosylated hemoglobin, cholesterol, white blood cell count, platelet, neutrophil count, lymphocyte count, eosinophil count, basophil count, monocyte count, neutrophil ratio, lymphocyte ratio, basophil ratio, and monocyte ratio.

### 3.2. The Risk of Recurrence of CRS in Children

To explore the risk factors associated with childhood recurrent sinusitis following FESS, we applied a binary logistic regression method for analysis. Univariate logistic regression revealed a negative association between systolic blood pressure and postoperative recurrence of CRS in children, as well as a positive association between a history of AR, asthma, and eosinophilic count and postoperative recurrence of CRS in children. The odds ratio of CRS recurrence decreased by 0.97 for every one-unit decrease in systolic blood pressure. Children with a history of AR had 10.45 odds of CRS recurrence compared to children without an AR history. Similarly, children with a history of asthma had 7.20 times the odds of CRS recurrence compared to those without a history of asthma. With every one-unit increase in eosinophil count, the odds of CRS recurrence increased by 59.69. For every unit increase in eosinophilic granulocyte ratio, the odds ratio for CRS recurrence was 1.32 (Table 2).

In all three models, children with CRS with a history of AR demonstrated an increased risk of postoperative recurrence compared to those CRS children without an AR history. This relationship remained essentially stable, suggesting that a history of AR is closely associated with an increased risk of postoperative CRS recurrence in children (*p* = 0.001). Specifically, we found that the risk of CRS recurrence was 21.04 times higher in children with a history of AR compared to those without, signifying that a history of AR represents an independent risk factor for the recurrence of CRS in children [odds ratio (OR): 22.04, 95% confidence interval (CI): 5.147–94.39] (Table 3).

## 4. Discussion

Children’s CRS is distinct from adult CRS and is a relatively understudied condition [5]. According to the existing international and domestic diagnosis and treatment guidelines, the symptoms of CRS in children often persist or relapse quickly and an effective method for the early diagnosis of CRS recurrence remains elusive [18]. Only children with poor responses to conventional treatments and who meet the surgical criteria were admitted for surgical intervention. Medical records were meticulously written, incorporating comprehensive medical histories and the latest biochemical indices pre-operation. Perioperative management was conducted in strict adherence to guidelines. Postoperatively, we continued with standardized drug treatment and regular outpatient reviews and evaluations. However, some participants still experienced unresolved symptoms even after FESS surgery. To identify potential risk factors, we followed up on children with CRS who had undergone surgery at our hospital in the past 5 years. Many previous studies combined adults and teenagers in all age groups and presented much shorter follow-up periods [19]. In this study, the longest follow-up time was 5 years, and the shortest follow-up time was 1 year. To our knowledge, our study represents one of the longest follow-up studies for children with CRS after FESS and one of the largest sample sizes compared to previous literature reports. All medical data and follow-up records of this study cohort are archived in a single healthcare provider database, thus facilitating the analysis of the disease continuum without losing data during follow-up [19].

The etiology of postoperative CRS recurrence is presently complex and unclear. It has been reported that patients with recurrent CRS respond less to maximum medication and surgery, and they have the highest risk of treatment failure and repeated recurrence. Thus, it is of significant clinical importance and clinical significance to clarify the factors related to the recurrence risk of CRS in children and evaluate their predictive value. Previous studies have indicated that asthma is a major risk factor for postoperative CRS recurrence in patients ranging from 11 to 60 years old [20]. P. C. Lu et al. suggested that Serum ECP appears to be a reliable predictor of early nasal polyp recurrence post-operation in a study involving participants aged between 36 to 57 years old (*n* = 58 cases) [21]. The E/M ratio [the threshold ratio of the total ethmoid sinus (E) and total maxillary (M) scores for both sides, determined by CT scanning as per the Lund-Mackay scoring system] has been reported as a highly accurate predictor of CRSwNP relapse, with a critical point of 2.55 indicating the highest predictive value of CRSwNP recurrence, as revealed in a study by Y. Meng et al. [22]. Prior studies reported that various factors such as mucus cystatin 2, pappalysin-A, periostin levels, Charcot-Leyden Crystal mRNA levels in nasal brushing or nasal secretions, high eosinophilic infiltration and high IL-5 expression, decreasing number of mast cells, occupational dust exposure and non-IgE-mediated asthma and blood eosinophil count combined with an asthma history were predictors of relapse of CRSwNP [23,24,25,26,27,28,29,30]. However, the aforementioned studies mostly involved participants aged 30–60 years old, and the sample size was less than 100. Currently, there are few studies on postoperative childhood recurrent sinusitis. In this study, all children with CRS or CRS recurrence were younger than 18 years old and were treated with standardized conservative drug treatments in outpatient or local clinics, strictly adhering to diagnosis and treatment standards before admission. The result of the research showed that the postoperative recurrence rate of children with AR history was significantly higher than that of children without AR [13 (48.51) vs. 8 (8.16), *p* < 0.000]. Allergic rhinitis is an important cause of sinusitis in children, and it is reported that about 20–80% of chronic rhinosinusitis in children is related to allergy [2]. AR is caused by an IgE-mediated response to inhaled allergens and is one of the most common chronic diseases in the world. AR-related allergens, including pollen, mold, and indoor allergens (house dust mites and animal allergens), vary widely within and between countries [9]. Allergens with protease activity can directly destroy the epithelial barrier or activate pattern recognition receptors and promote epithelial cells to release alarming proteins such as IL-33, thymic stromal lymphopoietin (TSLP), or IL-25 to initiate an innate immune response. Conversely, alarming proteins can activate group 2 innate lymphocytes (ILC2s), which rapidly produce in situ type 2 cytokines (IL-5, IL-13, and IL-4), resulting in mucosal inflammation [16]. Longitudinal reports have reported that the incidence of Th2 inflammatory CRSwNP has significantly increased in some regions of Asia. It was reported that mast cells were increased in number in CRS compared to control patients, producing type 2 cytokines by carrying IgE on their surface in type 2 immunity [9]. Cheng et al.’s recent study suggested higher absenteeism rates in schools among children and young people suffering from CRS [31]. Identifying the risk factors of postoperative childhood recurrent sinusitis could enable more targeted or personalized therapy, improving disease control and supporting them through critical school stages. So far, no research has reported that AR history is an independent risk factor for CRS in children. Multiple binary logistic regression was used to analyze the potential association between AR history and recurrence of sinusitis in children, and three models were constructed to confirm that the risk of recurrence of FESS in CRS children with AR undergoing FESS was 21.04 times higher than in those without AR history. This identifies AR history as an independent risk factor for CRS recurrence post-FESS in children. Understanding the influencing factors in children with CRS post-surgery is crucial for predicting postoperative prognosis. Allergic rhinitis may promote the recurrence of CRS in children post-FESS through several mechanisms. Allergic rhinitis can induce a hyper-responsive state in the nasal cavity and paranasal sinuses mucosa, leading to mucosal edema and mucus excretion disorders, which favors viral infection and bacterial proliferation. Anti-inflammation and anti-edema become the key points in the treatment of children’s sinusitis. Luo Zhang reported that the abundance of Campylobacter, associated with CRS recurrence, was positively correlated with IgE level [32]. This is in line with recent studies that link IgE level with the progression of allergic rhinitis from childhood into adulthood and with the rhinitis phenotype [33]. Increased IgE levels in AR patients may impact the distribution of nasal microflora, particularly encouraging the proliferation of Staphylococcus aureus, which can induce type 2 inflammation and promote CRS recurrence. In diagnosing upper airway inflammatory disease, other possible inflammatory conditions or medical history should be considered to ensure accurate diagnosis and reduce misdiagnosis.

Both AR and asthma were significant in both non-adjusted and adjusted models in this study.

The concept of “one airway, one disease” relies on epidemiological, clinical, functional, immunological, and histological relationships, leading to a global management approach for allergic respiratory diseases. There is also clear evidence that both eosinophils and macrophages play an important role in CRSwNP and CRSsNP, inducing type 2 inflammation and promoting CRS recurrence [34]. Recent advances in the use of biological agents to block the inflammatory molecular pathway of refractory CRSwNP offer new options for patients [34]. CRSwNP patients exhibiting Th2 inflammatory reactions and high serum IgE levels may benefit from biological agents, which have shown the advantages of high efficiency and safety in adult CRS clinical research and may be extended to pediatric patients in the future. For CRS in children accompanied by AR or asthma, proactive treatment is key to enhancing the therapeutic effect and reducing CRS recurrence.

This study’s findings suggest that newly diagnosed children with CRS undergo allergy testing and their family members should receive enhanced health education. Clinicians should bolster the comprehensive anti-inflammatory and anti-allergy treatment, pay close attention to the history of AR, simultaneously treat multiple diseases, and implement personalized treatment plans to improve disease prognosis. Children with CRS complicated by AR should be closely followed up post-FESS surgery, and efforts should be made to raise their awareness of AR control. These measures are likely to reduce the recurrence of CRS in children, improve the surgical outcomes of CRS and increase the success rate of surgical treatment.

In this study, it was also found that triglycerides and other blood lipids in serum also showed abnormal trends. The results suggest that abnormal blood lipid metabolism may also be a potential risk factor for the recurrence of sinusitis in children after an operation. Ursula Smole’s recent study suggested that the fatty acid binding protein (FABP) family promotes pulmonary type 2 immunity and is significantly up-regulated in chronic sinusitis [35]. In our study, the serum total protein levels were lower in recurrent CRS compared to initial CRS, and compared to those with initial CRS, the systolic blood pressure levels were relatively lower in children with recurrent CRS. These results may indicate some potential factors related to non-allergic ailments that need further study.

In this study, we used multiple logistic regression to quantify the independent actions of variables. This study is the first to examine children under 18 years old with CRS who had undergone FESS surgery at Xiangya Hospital, with the longest follow-up time reaching five years. The study analyzed the correlation between demographic characteristics, significant medical history, biochemical indexes, and the postoperative recurrence risk of CRS in children. The multivariate analysis shows that the model is stable and reliable, which can assist clinicians in considering simultaneous treatment of multiple diseases and predicting the prognosis of children with CRS who have a history of AR. Nevertheless, this study has limitations. First, this is a retrospective study investigating the relationship between AR, asthma history, demographic characteristics, and blood biochemical indexes of CRS children and the risk of postoperative recurrence. However, it does not establish a causal relationship. Secondly, the precise course of AR remains unclear, and some allergen tests were not performed in our hospital.

## 5. Conclusions

To the best of our knowledge, this is the first study to demonstrate that AR is an independent risk factor for postoperative childhood recurrent sinusitis. The findings of this study emphasize the importance of simultaneously treating AR in children with sinusitis, not only during their clinic visits but also in the perioperative period and post-operation, which is helpful for individualized treatment. It is necessary to conduct closer follow-up and reexamination of children with CRS and AR. There is a need to strengthen science education for parents and children to enhance their protective awareness and achieve better clinical outcomes. Intervention in allergic diseases can aid in the recovery from sinusitis in children, and biological agents may become one of the options for children with CRS.

## Figures and Tables

**Figure 1 children-10-01207-f001:**
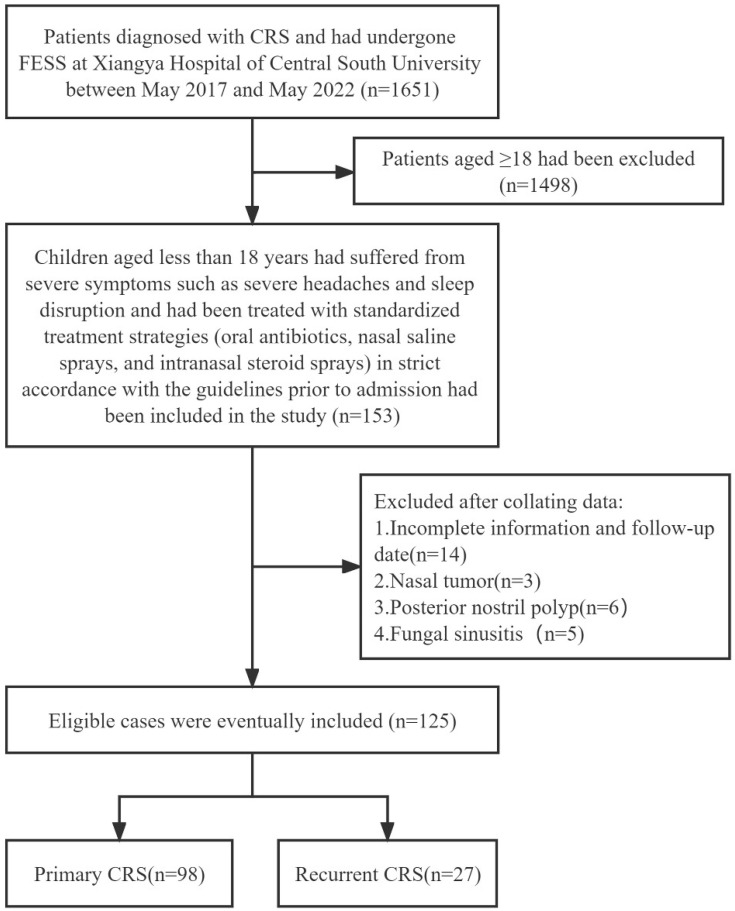
Inclusion and exclusion criteria.

**Table 1 children-10-01207-t001:** Comparison of general data of different outcomes in children with chronic sinusitis after FESS [*n* = 125, mean ± SD, *n* (%)].

Characteristics	Primary CRS (*n* = 98)	Recurrence CRS (*n* = 27)	*p*-Value
Sex			0.831
Male	71.00 (72.45)	19.00 (70.37)	
Female	27.00 (27.55)	8.00 (29.63)	
Age, years	15.00 (13.00–16.00)	15.00 (13.00–16.00)	0.792
Height, m	1.65 (1.55–1.72)	1.60 (1.49–1.74)	0.479
Weight, kg	54.69 ± 14.56	56.92 ± 17.35	0.501
Systolic blood pressure. mmHg	117.88 ± 13.71	112.00 ± 10.37	0.041
Diastolic pressure, mmHg	70.00 (63.50–78.50)	69.00 (61.00–77.00)	0.309
Course of disease. month	24.00 (12.00–48.00)	24.00 (12.00–48.00)	0.690
Allergic rhinitis			0.000
Yes	8.00 (8.16)	13.00 (48.51)	
No	90.00 (91.84)	14.00 (51.85)	
Asthma			0.012
Yes	3.00 (3.06)	5.00 (18.52)	
No	95.00 (96.94)	22.00 (81.48)	
Total protein, g/L	68.90 (66.20–72.25)	66.70 (65.20–70.50)	0.024
Albumin, g/L	43.00 (41.20–45.55)	41.90 (40.00–43.70)	0.108
Globulin, g/L	26.20 ± 3.19	25.06 ± 2.99	0.100
albumin-globulin ratio (%)	1.70 (1.50–1.80)	1.70 (1.50–1.90)	0.494
Total bilirubin, umol/L	8.50 (6.25–11.60)	9.80 (7.10–12.80)	0.279
Direct bilirubin, umol/L	3.70 (2.80–4.65)	4.10 (3.10–5.10)	0.640
Total bile acid, umol/L	4.00 (2.55–5.70)	3.50 (2.40–6.10)	0.688
Alanine aminotransferase, U/L	12.40 (9.55–16.95)	13.80 (9.90–24.10)	0.420
Glutamic oxalotransaminase, U/L	20.19 ± 4.81	20.24 ± 5.01	0.956
Urea, mmol/L	4.43 ± 1.07	4.35 ± 1.14	0.723
Creatinine, umol/L	68.00 (57.35–78.50)	67.00 (48.10–79.00)	0.597
uric acid, umol/L	373.30 (312.10–428.80)	382.40 (317.90–425.30)	0.723
Glucose, mmol/L	4.80 ± 0.57	4.64 ± 0.43	0.175
Glycosylated hemoglobin, mmol/L	2.05 (1.97–2.17)	2.05 (1.90–2.23)	0.662
Triglyceride, mmol/L	0.96 (0.73–1.14)	1.12 (0.88–1.59)	0.028
Cholesterol, mmol/L	3.67 (3.22–4.21)	3.99 (3.52–4.28)	0.082
HDL, mmol/L	1.17 (1.08–1.29)	1.17 (1.03–1.20)	0.395
LDL, mmol/L	2.17 (1.80–2.54)	2.42 (2.06–2.73)	0.067
Leukocyte count, 10^9^/L	6.15 (5.42–7.70)	6.00 (5.30–7.00)	0.408
Platelet, 10^9^/L	229.00 (197.00–260.00)	236.00 (186.00–278.00)	0.867
Neutrophil count, 10^9^/L	3.50 (2.50–4.40)	3.20 (2.80–3.60)	0.287
Lymphatic count, 10^9^/L	2.21 ± 0.66	2.27 ± 0.77	0.707
Eosinophilic count, 10^9^/L	0.10 (0.09–0.20)	0.10 (0.10–0.20)	0.226
Basophilic count, 10^9^/L	0.00 (0–0.04)	0.00 (0.00–0.03)	0.940
Monocyte count, 10^9^/L	0.50 (0.40–0.60)	0.40 (0.40–0.50)	0.402
Neutrophil ratio (%)	54.70 ± 10.33	52.19 ± 8.33	0.247
Lymphocyte ratio (%)	34.81 ± 8.96	36.77 ± 9.30	0.321
Basophil ratio (%)	0.50 (0.30–0.70)	0.60 (0.50–0.80)	0.075
Eosinophil ratio (%)	1.80 (0.95–3.00)	2.40 (1.20–4.60)	0.127
Monocyte ratio (%)	7.80 (6.10–9.87)	7.50 (6.70–8.10)	0.391

**Table 2 children-10-01207-t002:** Univariate binary logistic regression analysis of recurrence in children with CRS after FESS [*n* = 125, Mean ± SD, *n* (%)].

Characteristics	Statistics	OR	95%CI	*p*-Value
Sex		1.11	0.43–2.83	0.831
Male	90.00 (72.00)			
Female	35.00 (28.00)			
Age, years	15.00 (13.00–16.00)	0.98	0.84–1.15	0.834
Height, m	1.65 (1.54–1.72)	0.32	0.02–5.18	0.420
Weight, kg	55.09 ± 15.19	1.01	0.98–1.04	0.497
Systolic blood pressure. mmHg	116.58 ± 13.30	0.97	0.93–0.99	0.044
Diastolic pressure, mmHg	70.00 (63.00–78.00)	0.98	0.93–1.02	0.271
Course of disease. month	24.00 (12.00–48.00)	1.00	0.99–1.02	0.851
Allergic rhinitis		10.45	3.67–29.71	0.000
Yes	21.00 (16.80)			
No	104.00 (83.20)			
Asthma		7.20	1.60–32.40	0.010
Yes	8.00 (6.40)			
No	117.00 (93.60)			
Total protein, g/L	68.50 (65.85–71.83)	0.94	0.87–1.02	0.136
Albumin, g/L	42.85 (41.03–45.00)	0.88	0.76–1.03	0.104
Globulin, g/L	25.98 ± 3.17	0.89	0.77–1.02	0.102
White ball ratio (%)	1.70 (1.50–1.80)	1.36	0.26–7.06	0.711
Total bilirubin, umol/L	8.65 (6.40–11.65)	1.03	0.95–1.12	0.469
Direct bilirubin, umol/L	3.72 (2.80–4.90)	0.96	0.77–1.18	0.671
Total bile acid, umol/L	4.00 (2.50–5.83)	0.99	0.89–1.09	0.784
Alanine aminotransferase, U/L	12.40 (9.65–17.90)	1.01	0.97–1.05	0.671
Glutamic oxalotransaminase, U/L	20.17 ± 4.84	1.00	0.92–1.10	0.955
Urea, mmol/L	4.42 ± 1.08	0.93	0.62–1.39	0.721
Creatinine, umol/L	67.5 (55.45–78.75)	1.00	0.99–1.02	0.636
uric acid, umol/L	4.41 (3.67–5.05)	1.00	1.00–1.01	0.734
Glucose, mmol/L	4.76 ± 0.55	0.56	0.24–1.30	0.175
Glycosylated hemoglobin, mmol/L	2.05 (1.94–2.18)	0.42	0.05–3.85	0.440
Triglyceride, mmol/L	0.96 (0.76–1.23)	2.00	0.82–4.88	0.127
Cholesterol, mmol/L	3.80 (3.25–4.24)	1.31	0.77–2.21	0.320
HDL, mmol/L	1.17 (1.04–1.27)	0.34	0.04–2.93	0.326
LDL, mmol/L	2.33 (1.84–2.67)	1.31	0.72–2.39	0.374
Leukocyte count, 10^9^/L	6.13 (5.40–7.48)	0.87	0.67–1.15	0.336
Platelet, 10^9^/L	231.00 (195.25–261.75)	1.00	0.99–1.01	0.708
Neutrophil count, 10^9^/L	3.35 (2.60–4.20)	0.78	0.55–1.12	0.184
Lymphatic count, 10^9^/L	2.20 ± 0.64	1.13	0.61–2.09	0.704
Eosinophilic count, 10^9^/L	0.10 (0.10–0.20)	59.69	1.52–2344.94	0.029
Basophilic count, 10^9^/L	0.00 (0.00–0.03)	5.59	0–785,758.80	0.776
Monocyte count, 10^9^/L	0.50 (0.40–0.60)	0.35	0.03–3.99	0.395
Neutrophil ratio (%)	54.17 ± 10.00	0.97	0.93–1.02	0.247
Lymphocyte ratio (%)	35.26 ± 9.07	1.03	0.98–1.08	0.319
Basophil ratio (%)	0.50 (0.30–0.70)	2.28	0.87–6.01	0.095
Eosinophil ratio (%)	1.95 (1.00–3.18)	1.32	1.04–1.69	0.023
Monocyte ratio (%)	7.65 (6.23–9.78)	0.94	0.79–1.10	0.432

Adjusted for: none.

**Table 3 children-10-01207-t003:** Comparison of multiple multivariate binary Logistic regression for recurrence of CRS in children after FESS.

	OR	95%CI	*p*-Value
Model 1	10.45	3.67–29.71	0.000
Model 2	12.4	4.11–37.35	0.000
Model 3	22.04	5.147–94.39	0.000

Model 1: adjusted for none. Model 2: adjusted for sex, age, height and weight. Model 3: adjusted for sex, age, height, weight, serum total protein, serum globulin, white ball ratio, white blood cell count, neutrophil count, lymphocyte count, monocyte count, basophil ratio and eosinophil ratio.

## Data Availability

The data analyzed in this study are subject to the following permissions/restrictions: The original data supporting the conclusions of this paper will be provided by the author without any reservation. Requests to access these data sets should be sent directly to the corresponding author.

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
