# Peer review of "Allergic Rhinitis as an Independent Risk Factor for Postoperative Recurrence of Children Chronic Sinusitis"

_children, 2023, doi:10.3390/children10071207_

Round 1

Reviewer 1 Report

The authors present an interesting research on CRS  and the results are clearly presented, however, there are a few remarks and issues.

Abstract: Background part is actually the Aims and objectives of the study. I suggest adding a sentence or two on literature findings on CRS. Also,  I suggest to.omit the part on describing methodology (analyzing medical records) in this section, it shpuld be included in the Methods section. In the Aims section the authors should just mention they aimed at gaining better i sight into CRS in a retrospective observational childhood cohort.

I suggest renaming CRS as childhood recurrent sinusitis.

The authors need to include a paragraph in the Methods section of the manuscript describing follow-up, as there is nothing on folliw-up in this version.

I also suggest adding a number for the Ethivs Committee approval (number of approval and date).

Additionally, the.authors should debate more on the role of allergic diseases, since both AR and asthma were significant in both non-adjusted and adjusted models. And since total blood eosinophils were significant, too, thw authors should debate more on atopy and certain phenotypes (such ad T2- high phenotypes of allergy). The authors didn't describe the tests and procedures used for the establishment of AR and asthma, I suggest adding that, at least in the supplementary data. And the authors do not mention total serum IgE, it would be interesting to see whether IgE is significant, too, to debate more on allergy phenotypes.

Additionally, since triglicerids were significant in distinguishing the two groups of participants (and HDL nad LDL nearly significant), the authors should debte more.on this. Were these children obese/overweight? Rhe authirs analyzed body height and weight, but these are not relevant for children, as their height anf weight normally changes with age due to growth. The authors should include BMI.percentiles in the analysis instead of height and weight (preferably national curves for BMI percentiles, as these are.more accurate).and see whether these are significant, since obesity/overweightness are common comorbidites in asthma.and AR abd are known to aggravate symptoms.

Finally, the authors should describe the cohort better, perhaps add a table on the clinical characteristics of the cohort (age, gender, baseline values of certain  parameters, personal.and family history, especially on atopy and allergy etc.).

The manuscript should be edited by a native English speaker,.as.there are multuple errors and it is a bit difficult to understand certain parts of the manuscript due to.that

Author Response

Dear Professor,

Thanks for your letter and for reviewers' comments concerning our manuscript entitled "Allergic rhinitis as an independent risk factor for postoperative recurrence of children chronic sinusitis"(Manuscript ID: children-2464448). Those comments are all valuable and helpful for revising and improving our paper. Please see the attachment.

Reviewer 2 Report

Major comments

- Ln 77-79: please add some references to your statements.
- Ln 92-93: please elucidate on these "standardized drugs"
- Please add the protocol number (or equivalent) for the Ethics Committee's approval
- The Authors note a relationship between certain clinical parameters and recurrence of CRS, namely blood pressure and serum tryglicerides (both increased) and serum protein levels (decreased). These results are stated but not discussed at all, and could be related to non-allergic ailments: point in case, serum protein levels are increased in allergic diseases, whereas blood tryglicerides can be traced to incorrect dietary patterns and/or altered lipid metabolism, and blood pressure itself is a rather generic marker in a multitude of cardiovascular diseases.
- In this study, he Authors confirm the role of asthma and allergic rhinitis as predictors of relapse in CRS. Were these ailments investigated? E.g. specific sensitizations, disease severity and current treatment regimens? I would consider this to be crucial, in order to stratify relapse risk and offer guidance to the clinician. 

Minor comments

- Figure 1: typo, should be "data"
- Ln 124: I assume the Authors meant "mean and interquartile ranges"? Furthermore, "categorical" in place of "classified
- Ln 133: "white ball ratio"?
- Ln 221: "E/M ratio" should be described, rather than assume the reader is going to refer to the original paper

I suggest a rather vigorous revision of the text by a native English speaker, as there are typos, convoluted sentences (e.g. Ln 48-50, 81-82, and others) and cases of incorrect syntax thoughout the entire paper. The Abstract is particularly egregious in this sense, possibly due to length constraints. 

Author Response

(The authors gave the same response as above.)

Reviewer 3 Report

Thank you for considering the changes to the manuscript.

While there has been considerable improvement, there are still a few items that we would like to see considered.

1. In Table 1, Some items are considered to be indicated by maximum-minimum or 95% CI values( Age etc.,), but no explanation is given for them. I hope that this will be revised as it is difficult to understand.

2. Is this study registered in the Clinical Trials Database? If so, please provide the registration number.

3. Please describe the limitations and generalizability obtained from this study in the discussion section.

Author Response

(The authors gave the same response as above.)

Round 2

Reviewer 2 Report

We thank the Authors for their revised text, addressing the concerns raised in the previous review round.

Ln 106-110: still markedly unclear and convoluted, with frequent repetitions.
Ln 144: "normal distribution"
Ln 219-228: unclearly worded. The "Discussion" section overall still needs to be revised, language-wise